# Constitutive Model for Thermal-Oxygen-Aged EPDM Rubber Based on the Arrhenius Law

**DOI:** 10.3390/polym16182608

**Published:** 2024-09-14

**Authors:** Xiaoling Hu, Xing Yang, Xi Jiang, Kui Song

**Affiliations:** 1School of Mechanical Engineering and Mechanics, Xiangtan University, Xiangtan 411105, Chinasongkui@xtu.edu.cn (K.S.); 2Hunan Key Laboratory of Geomechanics and Engineering Safety, Xiangtan University, Xiangtan 411105, China

**Keywords:** thermal oxygen aging, constitutive model, Arrhenius, mechanical behavior

## Abstract

Ethylene-propylene-diene monomer (EPDM) is a key engineering material; its mechanical characterization is important for the safe use of the material. In this paper, the coupled effects of thermal degradation temperature and time on the tensile mechanical behavior of EPDM rubber were investigated. The tensile stress-strain curves of the aged and unaged EPDM rubber show strong nonlinearity, demonstrating especially rapid stiffening as the strain increases under small deformation. The popular Mooney–Rivlin and Ogden (*N* = 3) models were chosen to fit the test data, and the results indicate that neither of the classical models can accurately describe the tensile mechanical behavior of this rubber. Six hyperelastic constitutive models, which are excellent for rubber with highly nonlinearity, were employed, and their abilities to reproduce the stress-strain curve of the unaged EPDM were assessed. Finally, the Davis–De–Thomas model was found to be an appropriate hyperelastic model for EPDM rubber. A Dakin-type kinetic relationship was employed to describe the relationships between the model parameters and aging temperature and time, and, combined with the Arrhenius law, a thermal aging constitutive model for EPDM rubber was established. The ability of the proposed model was checked by independent testing data. In the moderate strain range of 200%, the errors remained below 10%. The maximum errors of the prediction results at 85 °C for 4 days and 100 °C for 2 and 4 days were computed to be 17.06%, 17.51% and 19.77%, respectively. This work develops a theoretical approach to predicting the mechanical behavior of rubber material that has suffered thermal aging; this approach is helpful in determining the safe long-term use of the material.

## 1. Introduction

Ethylene-propylene-diene monomer (EPDM) rubber shows excellent heat resistance and outstanding elasticity and chemical stability, and it is widely used in aerospace, automobiles, high-speed trains and so on. For example, EPDM rubber has been used as the white side walls of tires and outer windshields of high-speed trains. In these applications, EPDM rubbers are often exposed to oxygen and elevated temperatures for decades. The long-term effects of thermal oxygen inevitably lead to microstructural changes in the rubber material, which ultimately cause degradation in the macroscopic mechanical properties [1,2,3,4]. Therefore, thermal oxygen aging in EPDM rubber could be a major concern in various industries, as it impacts the safety of the material for its long-term use.

Thermal oxygen aging in rubber is thermally activated; therefore, an elevated temperature may accelerate the aging process. Many studies of thermal oxygen aging in rubber were focused on determining property deterioration and lifetime predictions by conducting accelerated aging experiments. The accelerated aging experimental data were further extrapolated using the Arrhenius law or time-temperature principle (TTP) to predict the materials’ behavior for long-term use [2,3,4,5,6]. Thermal oxygen aging makes the characterization of the rubber more challenging, and thus, establishing a suitable constitutive model to predict the variation in the mechanical performance of rubber subjected to long-term thermal conditions is imperative for engineering applications of rubber material.

Many constitutive approaches have been developed to predict the mechanical responses of aged rubber. Numerous models have been constructed based on the idea of thermal oxygen aging in rubber caused by chemical reactions [7,8]. Johlitz et al. developed a thermo-mechanical coupled model by introducing two internal variables, *q*_d_ and *q*_r_, which represent the network degradation and network reformation process, respectively [9]. However, this model contains more than ten parameters and is limited to small deformations. Maryam Shakiba et al. present a physics-based constitutive equation, which was obtained by modifying the network stiffness and chain extensibility in the well-known Arruda-Boyce constitutive equation with the cross-link density evolution of elastomers during thermochemical aging [10]. This model is much simpler, but the model was developed based only on the Arruda-Boyce constitutive equation, and its accretion is dependent on the cross-linking density test results. Mohammadi proposed a micro-mechanical model to describe the quasi-static mechanical response and Mullins effect of elastomers that have suffered thermal aging. The model can capture the thermal-induced aging effect and can be determined easily, but it is also limited to a small deformation range [11]. Other proposed models based on the aging mechanism can be found in the literature [10,12,13,14,15,16]. Although these models can capture the thermal-aging-induced mechanical behavior of rubber material, they are dependent on an additional chemical characterization test.

As is well known, as temperature increases, the chemical reaction rate follows the Arrhenius law. Therefore, phenomenological models that are used to capture the constitutive behavior of aging rubber can be constructed according to the Arrhenius function. Khanh et al. [5] studied the effect of thermal aging on variations of the Mooney–Rivlin constants *C*_1_ and *C*_2_ applied to a polychloroprene rubber (CR), and the dependence of the elastic constants *C*_1_ and *C*_2_ on aging time and aging temperature were found to follow the Arrhenius relationship. Liu et al. assumed that the permanent compression strain of EPDM rubber obeys the Arrhenius law, and a stress-hydrothermal aging model combined with the Mooney–Rivlin equation was developed [17]. The developed model can successfully predict the time-dependent characteristics of the mechanical properties of rubber under long-term compressive stress. Recently, Mohammadi et al. used three to six parameters for an Arrhenius-based aging decay function, and a model to fit the aged rubber experimental observations was constructed [18]. Lou et al. employed two types of exponential models to describe the dependence of second-order classical Ogden hyperfoam model parameters on temperature and aging time [19]. The indexes of the exponential models are defined as the kinetic rate parameters related to the temperature, which follows the Arrhenius law. They finally developed an accurate model to describe the mechanical response of foams that suffer compressive strains under the long-term influence of high temperatures. The exponential model is an empirical formula that is dependent on parameter variation results without rigorous theoretical derivation.

In this study, the coupled effects of thermal degradation temperature and time on the tensile mechanical behavior of EPDM rubber were investigated. An accurate hyperelastic model for the tensile behavior of the rubber was determined. By analyzing the dependence of the model parameters on aging time and aging temperature, a constitutive model considering thermal-oxidative aging was constructed based on the Arrhenius law. Furthermore, independent testing data were obtained to assess the proposed model. Rubber material is inevitably subjected to thermal-oxidative aging during service, leading to severe mechanical performance degradation. This study provides a new approach for constructing a constitutive model for thermal-oxygen-aged rubber, which is of great significance for the long-term safe use of materials.

## 2. Materials and Methods

### 2.1. Materials and Specimen

The EPDM rubber used in this study was sheet rubber, 2 mm in thickness, supplied by Zhuzhou Times New Material Technology Co., Ltd., Zhuzhou, China. The chemical structure of the EPDM rubber is shown in Figure 1. The main formulation of the EPDM rubber compound is as follows: 100 phr EPDM, 30 phr Silica, 10 phr Zinc oxide, 2 phr Stearic acid, 50 phr Aluminum hydroxide, 20 phr Titanium dioxide, 8 phr Paraffin oil, 5 phr Antioxidant, 6.4 phr Vulcanization activator. Finally, the sheets were processed into dumb-bell specimens according to ISO 37-2011 [20] for the aging tests, and the size of the specimen is shown in Figure 2.

### 2.2. Experiment Setup

To investigate the coupled effects of thermal degradation temperature and thermal degradation time on the EPDM rubber, accelerated thermal aging experiments were carried out in a thermal aging chamber with an accuracy of ±1 °C. The specimens were aged at 70 °C, 85 °C and 100 °C for a total period of 84 days and then removed from the thermal aging chamber at specific time points (7 days, 14 days, 28 days, 56 days, 77 days and 84 days). At least three specimens were tested under each aging condition. After reaching the prescribed aging time, the specimens were shelved in air for more than 16 h. Then, uniaxial tensile tests were performed to evaluate the change in the mechanical properties of EPDM rubber after aging. The uniaxial tensile tests were carried out using an AGS-X electronic testing machine with a tensile rate accuracy of ±0.1%, and all tests were displacement controlled at a tensile rate of 200 mm/min till the tensile strain was up to 500%. Finally, the median of the three repeated samples was taken as the stress-strain curve of the material

### 2.3. Test Results

Figure 3 shows the tensile stress-strain curves of the EPDM rubber before and after aging at 70 °C, 85 °C and 100 °C for different aging days, respectively. It can be observed that the stress increased rapidly at the initial stage and then increased slowly, finally rapidly increasing again when the strain exceeded a certain value. The tensile behavior of the EPDM rubber shows strong nonlinearity, especially demonstrating rapid stiffening as the strain increased under small deformation, which is similar to some highly filled rubber. In addition, it can be found that as the temperature increases, the slope of the stress-strain curves increases slightly for small tensile strain, while a significant slope increase of the curves is observed when the tensile strain is large. At a temperature of 70 °C aged for 84 days, the stress at 500% strain is 8.52 MPa, and at 85 °C aged for 84 days, the value is 10.19 MPa. When the temperature reaches 100 °C aged for 77 days, the stress at the 500% strain reaches 13.27 MPa, which is about 1.6 times the value at 70 °C aged for 84 days. This shows that the temperature enhances the strength at a certain level of strain and results in strong nonlinearity of the material. All these observed phenomena demonstrate that the temperature and aging time both play an important role in the variation in mechanical properties.

## 3. Thermo-Oxidative Aging Constitutive Model for EPDM Rubber

### 3.1. Constitutive Relation

For rubber material, its hyperelastic properties can be described in terms of a strain energy function (W). The strain energy functions can be written in a polynomial form composed of strain invariants,
(1)$$W=W(I_{1},I_{2},I_{3})$$
where $I_{1}$, $I_{2}$ and $I_{3}$ are the three strain invariants of the Green deformation tensor. They are given in terms of the principal extension ratios $\lambda_{1}$, $\lambda_{2}$ and $\lambda_{3}$.
(2)$$\left\{\begin{array}{l} I_{1}=\lambda_{1}^{2}+\lambda_{2}^{2}+\lambda_{3}^{2}\\ I_{2}=\lambda_{1}^{2}\lambda_{2}^{2}+\lambda_{1}^{2}\lambda_{3}^{2}+\\ I_{3}=\lambda_{1}^{2}\lambda_{2}^{2}\lambda_{3}^{2} \end{array}\right.\lambda_{2}^{2}\lambda_{3}^{2}$$

The engineering (or nominal) stress *P* can be obtained as follows:(3)$$P_{i}=\left[\frac{\delta W}{\delta I_{1}}\frac{\delta I_{1}}{\delta \lambda_{i}}+\frac{\delta W}{\delta I_{2}}\frac{\delta I_{2}}{\delta \lambda_{i}}+\frac{\delta W}{\delta I_{3}}\frac{\delta I_{3}}{\delta \lambda_{i}}\right]+\frac{P}{\lambda_{i}}$$

For the case of homogeneous, isotropic and incompressible or nearly incompressible hyperelastic rubber, $I_{3}=1$. In the case of uniaxial tension, the loading stretch along direction 1 can be redefined as $\lambda_{1}=\lambda =1+\varepsilon$, with e being the strain in the tensile direction. For the other two transverse directions, $\lambda_{2}=\lambda_{3}=\lambda^{-1/2}$. The engineering (or nominal) stress *P* under the uniaxial tensile condition can be written as
(4)$$P_{1}=2\left(\frac{\partial W}{\partial I_{1}}+\frac{1}{\lambda }\frac{\partial W}{\partial I_{2}}\right)\left(\lambda -\frac{1}{\lambda^{2}}\right)$$

### 3.2. Hyperelastic Constitutive Model for EPDM Rubber

The Mooney–Rivlin constitutive model is one of the most widely used models for EPDM [21], and another widely used model to describe the hyperelastic response of rubber material is the Ogden model of order *N* = 3 [22]. However, in the study of He et al. [23], they found that many classical hyperelastic constitutive models failed to capture a specific mechanical property, which is that rubber materials soften under small deformation but rapidly stiffen as the strain increases. For this reason, He et al. evaluated 85 types of hyperelastic models for different rubber materials and proposed 6 top models to reproduce this specific mechanical response. And the top six models are Davis–De–Thomas [24], modified Yeoh [25], Alexander [26], gen-Yeoh [27], Gregory [23] and modified Gregory [28]. In the previous section, we found that the EPDM rubber studied in this work shows the same features in the small deformation range as some highly rubber material. Due to the tension strain range reaching 500% in our study, which is larger than the 100% strain range in He’s study, the applicability of the six top models for our test data is not clear. In the following, we will select the most accurate model for the EPDM rubber by fitting the models with the test data. The stress and elongation relationships of those models are listed in Table 1.

### 3.3. Evaluation of Models for EPDM Rubber

The Mooney–Rivlin model and Ogden model of order *N* = 3 were used to fit the experimental stress-strain data of the unaged rubber, as shown in Figure 4. It can be seen that there is significant deviation between the fitting result of the Mooney–Rivlin model and the tensile test data, and the model cannot fit the “S-shape” characteristics of the stress-strain curve under large deformation, which has been found in many previous studies [5,29]. In Figure 4, we can see that although the Ogden model of order *N* = 3 can reflect the main characteristics of the tensile behavior in a large deformation, there is still an obvious deviation between the fitting curve and test data in the small deformation range. The Ogden model is successful in capturing the nonlinear hyperelastic behavior of many kinds of rubbers, so why does it not fit the data in Figure 4 well? This may be due to the fact that the stress increases rapidly as strain increases at a low strain range, and the stress-strain curve shows a more prominent nonlinear characteristic.

To check the applicability of the six top models, we fitted them to the test data in Figure 4, and the results are shown in Figure 5. It can be seen that the Davis–De–Thomas model, modified Gregory model and Gregory model can perfectly model the stress-strain data of the unaged EPDM, while the modified Yeoh model, Alexander model and Gen-Yeoh model do not. From Figure 5, we also can conclude that although the six models are successful in reproducing the stress-strain curve with strong nonlinear characteristics in the 100% strain range, when the strain is large those models are not all suitable. Models with fewer parameters and considerable modeling ability are considered to be the best models. Thus, the Davis–De–Thomas model with four parameters was selected as the constitutive model of the EPDM rubber.

### 3.4. Effect of Temperature and Aging Time on Model Parameters

To construct the thermal aging constitutive model for the EPDM rubber, the relationships between model parameters and temperature and aging time should be established. By fitting stress-strain data under different aging temperatures and aging times, the four parameters, *m*, *n*, *c* and *k*, of the model as functions of temperature and aging time were obtained, as shown in Figure 6. In Figure 6, we can see that the parameters *m*, *n* and *k* increase gradually with increasing aging temperature and aging time, while parameter *c* shows an opposite trend. In addition, all of the parameters vary with a time approximated exponential relationship under different aging temperatures.

For the thermal aging process, the rate of thermal degradation satisfies a Dakin-type kinetic relationship [30]:(5)$$\frac{dx}{dt}=kf\left(x\right)$$
where x is the investigated property, *t* is the thermal aging time, *k* is the kinetic rate and $f\left(x\right)$ is a function of the degree of material degradation. The kinetic rate *k*, related to the temperature, is assumed to follow the Arrhenius law:(6)$$k=A\exp\left(-\frac{E_{a}}{RT}\right)$$
where A is the pre-exponential factor, $E_{a}$ the activation energy, *R* the gas constant and T the absolute temperature. As shown in Figure 6, all of the parameters caused by thermal aging follow an exponential relation with aging time. The change in parameters with aging time and aging temperature is defined as:(7)$$f\left(x\right)=x^{n}$$
in which *n* is the parameter that needs to be determined by fitting.

Combining the above three equations, the relationship functions between the four parameters and temperature and aging time can be written uniformly as
(8)$$x={\left(\left(1-n\right)A\exp\left(-\frac{E_{a}}{RT}\right)t+{x_{0}}^{1-n}\right)}^{\frac{1}{1-n}}$$
where $x_{0}$ corresponds to the model parameters of the unaged rubber. The activation energy value is 51.1 kJ/mol, which was identified in our previous study [31].

Equation (8) was used to fit the data in Figure 6 by the nonlinear least-squares method. Here, we employed the nonlinear surface fitting approach, which can fit parameters *m*, *n*, *c* and *k* of the three curves under three temperatures simultaneously. The fitting curves are shown in Figure 6. The R-square values for the aging model parameters *m*, *n*, *c* and *k* are 0.9679, 0.9691, 0.969 and 0.9059, respectively. Thus, the temperature- and aging time-dependent model parameters were established as Equation (9). Finally, the thermal aging model for the thermal-aged EPDM rubber can be expressed by combining the Davis–De–Thomas model and Equation (9).
(9)$$\left\{\begin{array}{l} m={\left(2.19E7\cdot e^{-\frac{6194.37}{T}}\cdot t+4.34\right)}^{0.13}\\ n={\left(2.16E5\cdot e^{-\frac{6194.37}{T}}\cdot t+2.09\right)}^{-0.85}\\ c={\left(2.82\cdot e^{-\frac{6194.37}{T}}\cdot t+1.23E-7\right)}^{0.26}\\ k={\left(0.15\cdot e^{-\frac{6194.37}{T}}\cdot t+2.70E-8\right)}^{0.32} \end{array}\right.$$

### 3.5. Model Verification

A thermal aging model of EPDM was developed by correlating the parameters in the Davis–De–Thomas model with aging temperature and aging time. In order to verify the validity of this model, the stress-strain data of samples aged at 70 °C for 7 and 28 days, 85 °C for 7 and 14 days and 100 °C for 7 and 56 days were used. Figure 7 shows the comparison of experimental and prediction curves, and the results show that the model can reproduce the aged stress-strain data at the three aging times of 14, 28 and 56 days, while the model fitting results obviously deviate from the test data of the samples aged for 7 days.

To further check the predictive ability of the thermal aging model, independent data should be used. Tensile stress-strain data for EPDM rubber aged at 85 °C for 4 days and 100 °C for 2 and 4 days were obtained by conducting new experiments on the EPDM rubber according to the methods in the Experiment Setup Section. The test data compared to the predicted results of the thermal-aging model are shown in Figure 8. It can be found that the model prediction results are in agreement with the actual stress-strain data in the moderate strain range of 200%. As the strain increases, the response for the EPDM predicted by the proposed model underestimates the experimental observations. Figure 9 shows the average prediction errors of the proposed model at a constant strain of 100%, 200%, 300%, 300%, 400% and 500%, which is defined as (test data-model prediction)/test data × 100%. In the moderate strain range of 200%, the errors remain below 10%. Furthermore, the maximum errors of the prediction results at 85 °C for 4 days and 100 °C for 2 and 4 days were computed to be 17.06%, 17.51% and 19.77%, respectively.

The developed aging model is based on the relationship between the parameters of the Davis–De–Thomas model and aging time/temperature. Therefore, the accuracy of the relationship is particularly important for the prediction effect of the aging model. Based on the rubber constitutive model, we analyzed the large deviation between the experimental results and the model results under the condition of short aging time and large deformation. Assuming that the strain *ε* approaches 0 (i.e., the tensile ratio *λ* approached 1), the results of the expressions *λ*^2^ + 2/*λ* − 3 and *λ* − *λ*^−2^ in the model will approach 0. Now for analysis, the result of can be assumed to be a very small amount *ξ*, then, the stress given by the mode can be estimated as *σ* ~ *ξmc*^-*n*^. So, at very small deformation, *σ* is only related to the parameters *m*, *n* and *c* and is almost unaffected by *k*. However, when deformation (tensile ratio *λ*) gradually increases, the coefficient of *k* given by the model is 4(*λ*^2^ + 2/*λ* − 3)(*λ* − *λ*^−2^). Therefore, with the increase in deformation, the stress *σ* of the model will gradually increase under the influence of *k*. Based on the fitting results of the four parameters, the R-square values of parameters *m*, *n* and *c* are 0.9679, 0.9694 and 0.969, respectively, while the R-square value of parameter *k* is the lowest one, at only 0.9059. As can be seen in Figure 6 (for parameter *k*), the relative fitting deviation under a short aging time is larger than that under a long aging time, which is the reason that the model has a large deviation in the large strain range when predicting the stress-strain curve with a short aging time. Although our model deviates from the experimental results during large deformations, they remain within a reasonable low-limit error range. It can be argued that the developed thermal aging model yields acceptable predictions even when only a simple tensile test is utilized.

## 4. Conclusions

The coupled effects of aging temperature and time on the tensile behavior of EPDM rubber were investigated. The test results indicate that the EPDM has strong nonlinear characteristics, especially at a small strain range. Moreover, aging temperature and time can enhance the nonlinearity of the material. Due to the high nonlinearity of the EPDM rubber, the classical Ogden model of order 3 and the Mooney–Rivlin model cannot accurately describe the experimental stress-strain data. Six top models for highly nonlinear rubber were employed to reproduce the tensile stress-strain curve of the unaged EPDM, and it was shown that not all of the six top models could capture the stress-strain in a strain range of 500%, and that the Davis–De–Thomas model was the best one. The Davis–De–Thomas model was used to fit the test data of the EPDM rubber aged under different aging temperatures and times, and the model parameters were found to vary exponentially with the increasing aging time. A thermal aging constitutive model for the EPDM rubber was established by employing the Arrhenius law and Dakin-type kinetic relationship to describe the relationships between the aging model parameters and aging temperature and time. Moreover, independent testing data were used to verify the developed model. Results show that the developed model can describe the thermal aging mechanical responses of the EPDM, especially when the tensile strain is in the range of 200%.

## Figures and Tables

**Figure 1 polymers-16-02608-f001:**
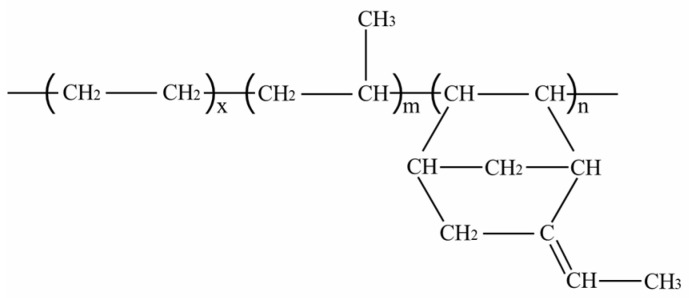
Chemical structure of the EPDM rubber.

**Figure 2 polymers-16-02608-f002:**
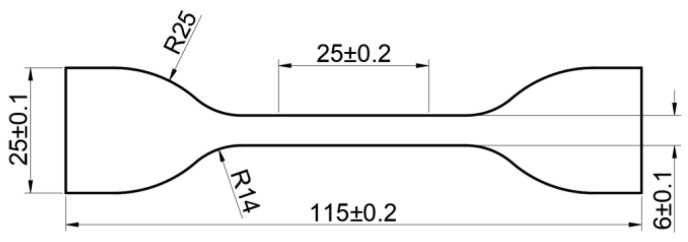
Size of specimens (unit: mm).

**Figure 3 polymers-16-02608-f003:**
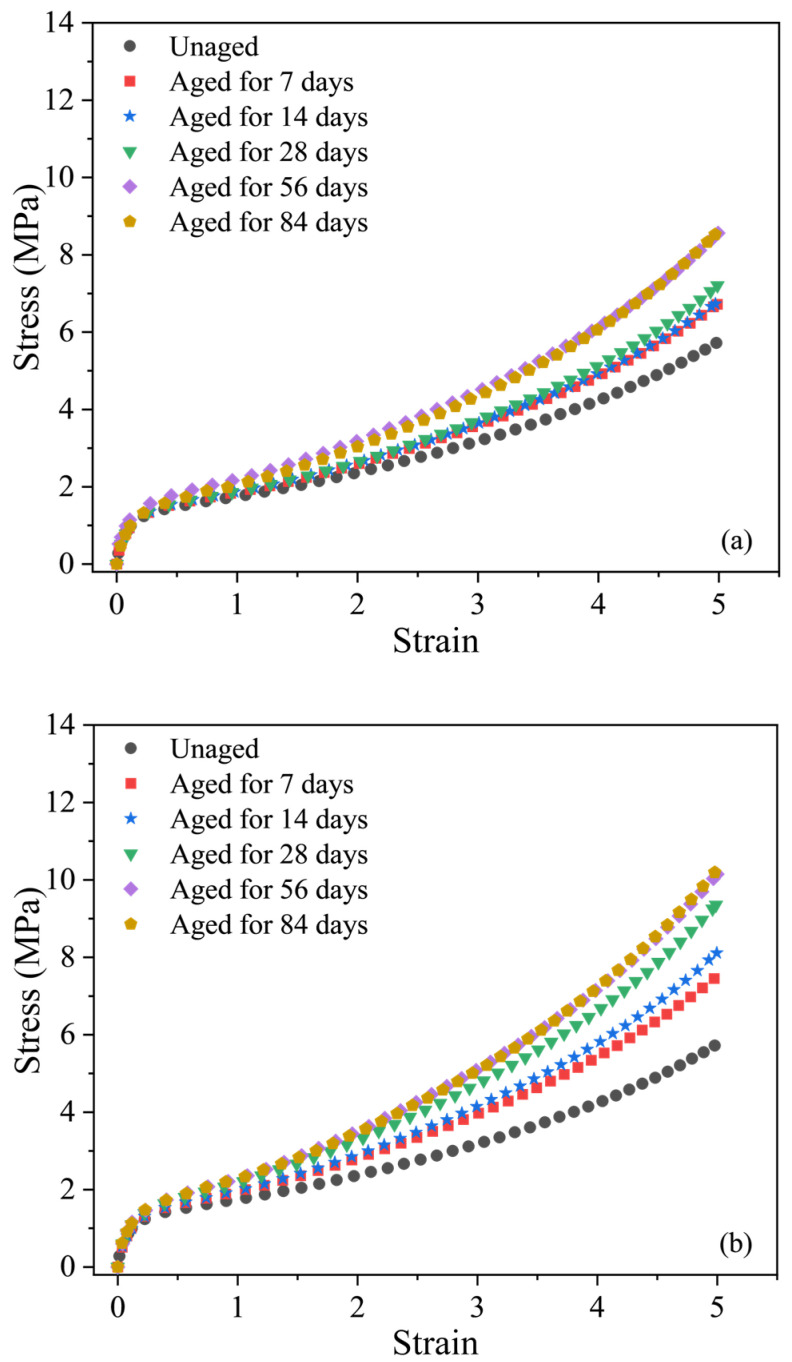
Stress-strain curves of the EPDM before and after aging for different times at (**a**) 70 °C; (**b**) 85 °C; (**c**) 100 °C.

**Figure 4 polymers-16-02608-f004:**
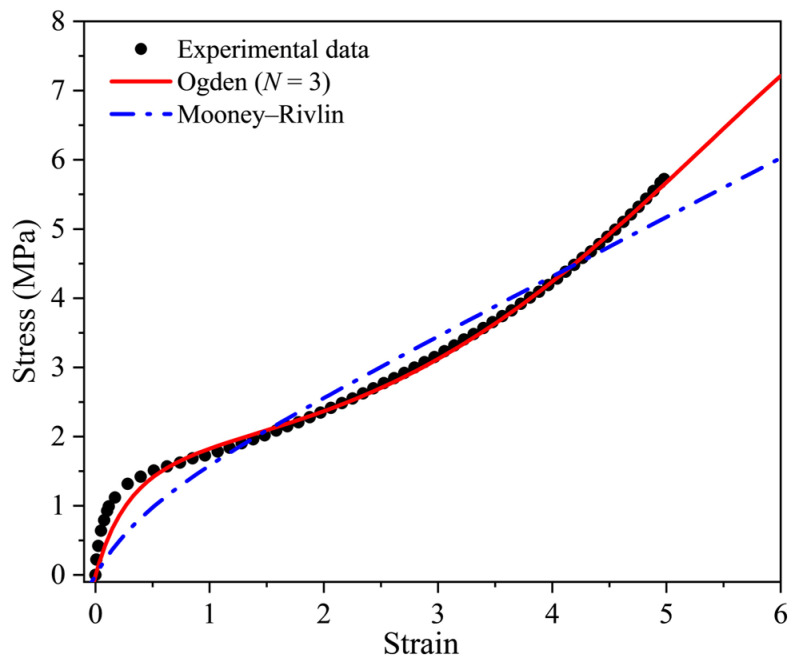
Fitting curves vs. the stress-strain data of the unaged EPDM rubber by the two classical models.

**Figure 5 polymers-16-02608-f005:**
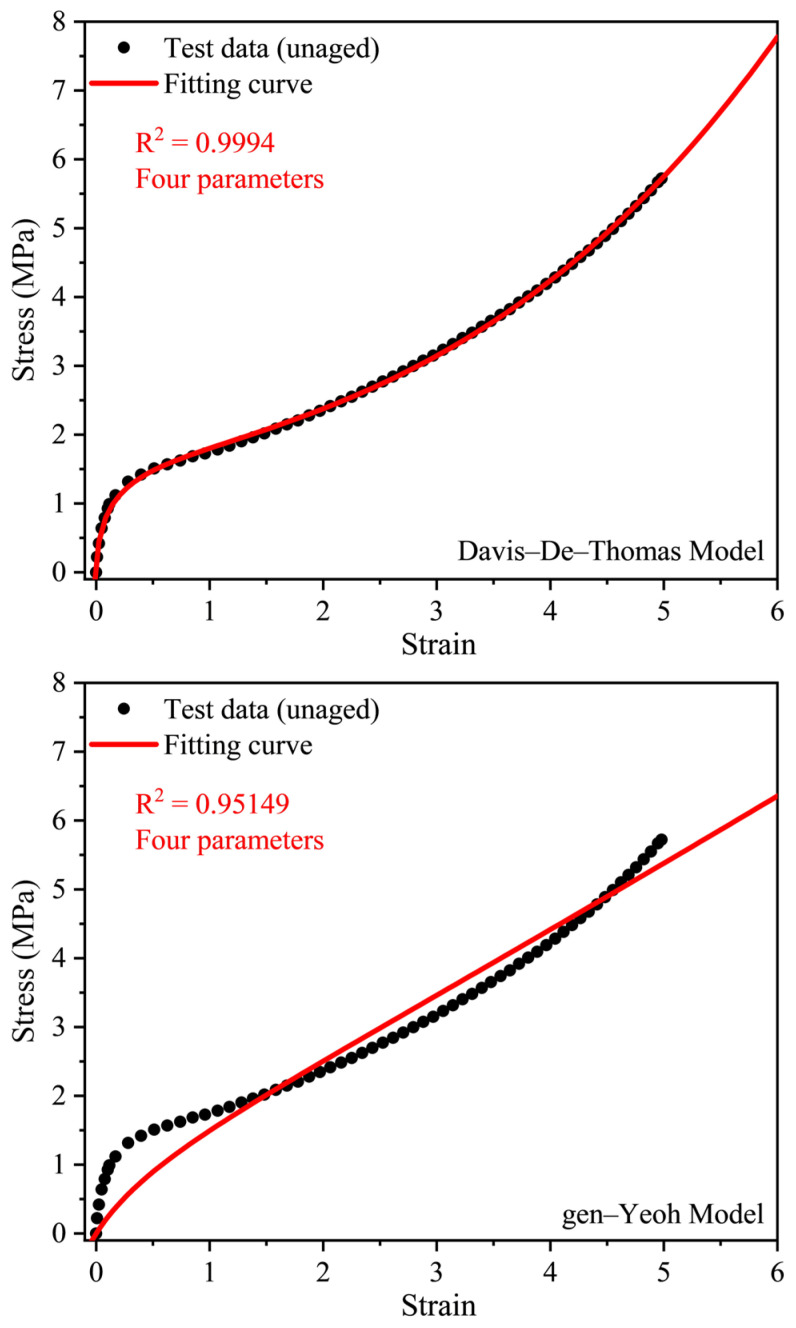
Fitting curves vs. the stress-strain data of the unaged EPDM rubber by the top six models.

**Figure 6 polymers-16-02608-f006:**
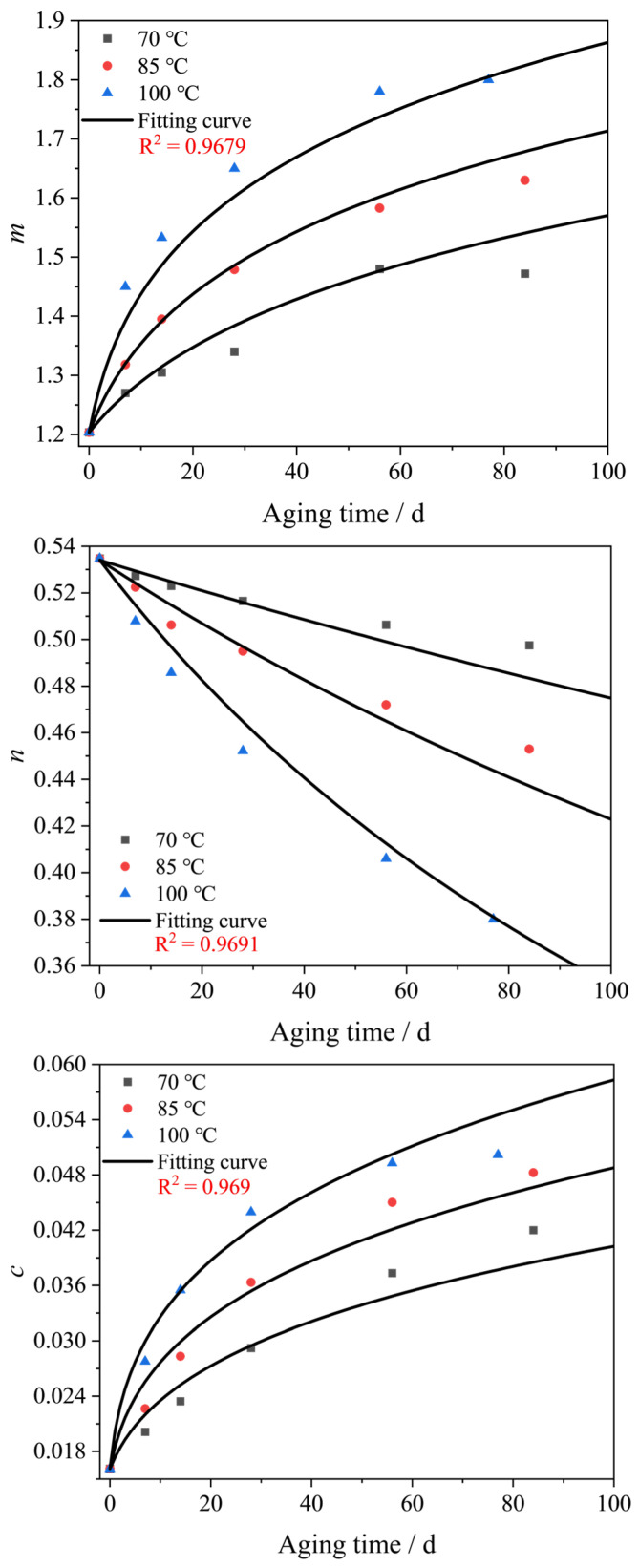
Plots of model parameters *m*, *n*, *c* and *k* vs. aging time under different aging temperatures.

**Figure 7 polymers-16-02608-f007:**
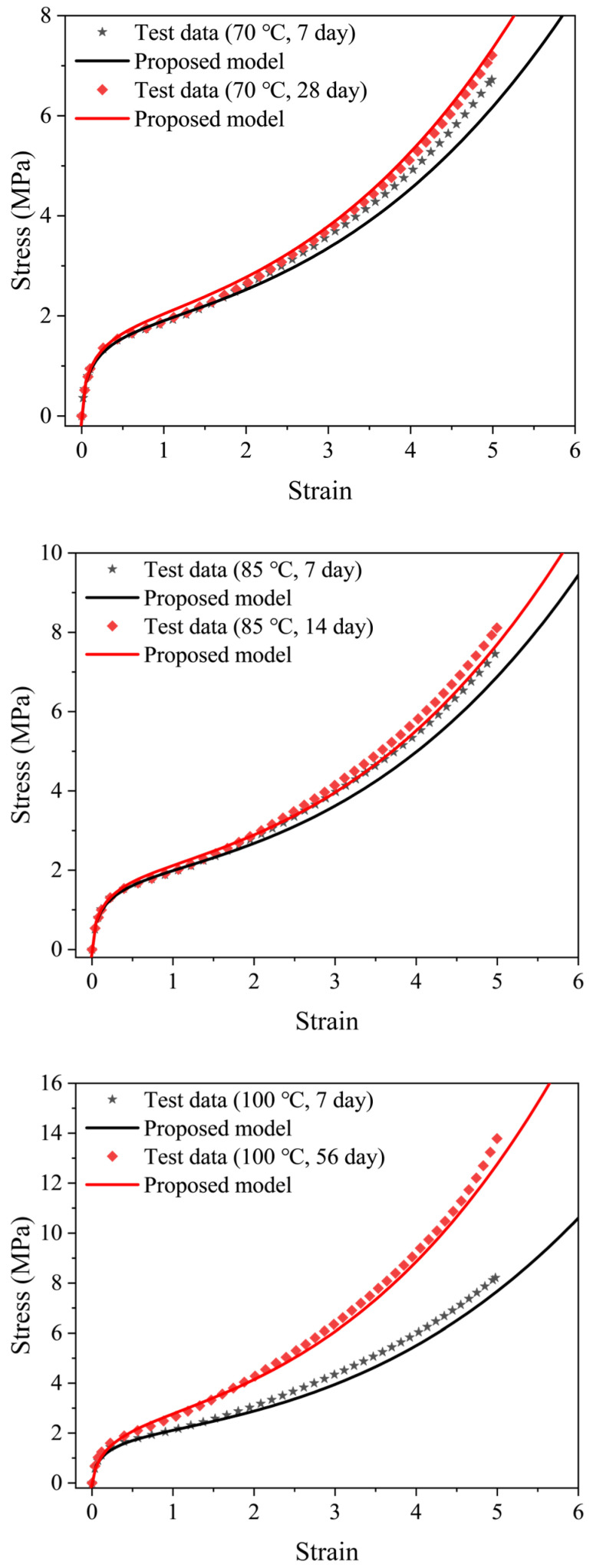
Comparison between the prediction results of the developed model and the test stress-strain data with EPDM aged at 70 °C for 28 days, 85 °C for 14 days and 100 °C for 56 days.

**Figure 8 polymers-16-02608-f008:**
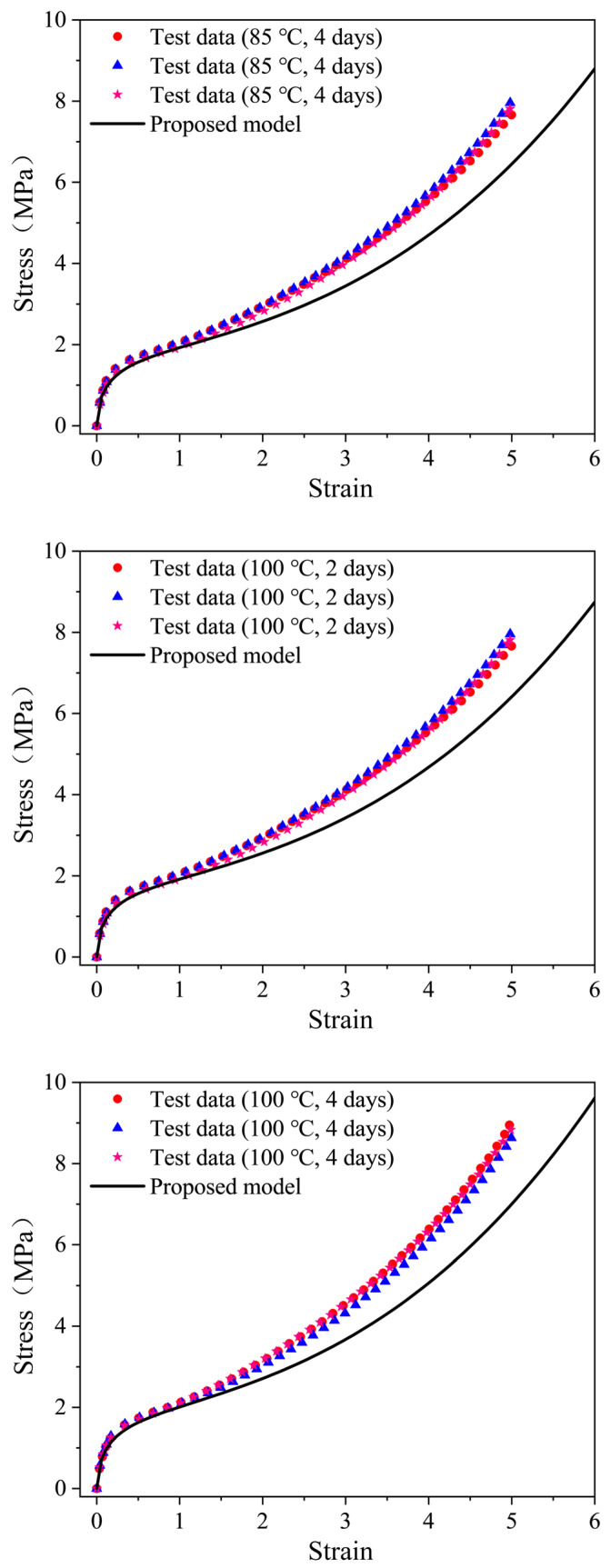
Comparison between the prediction results of the developed model and the independent test stress-strain data.

**Figure 9 polymers-16-02608-f009:**
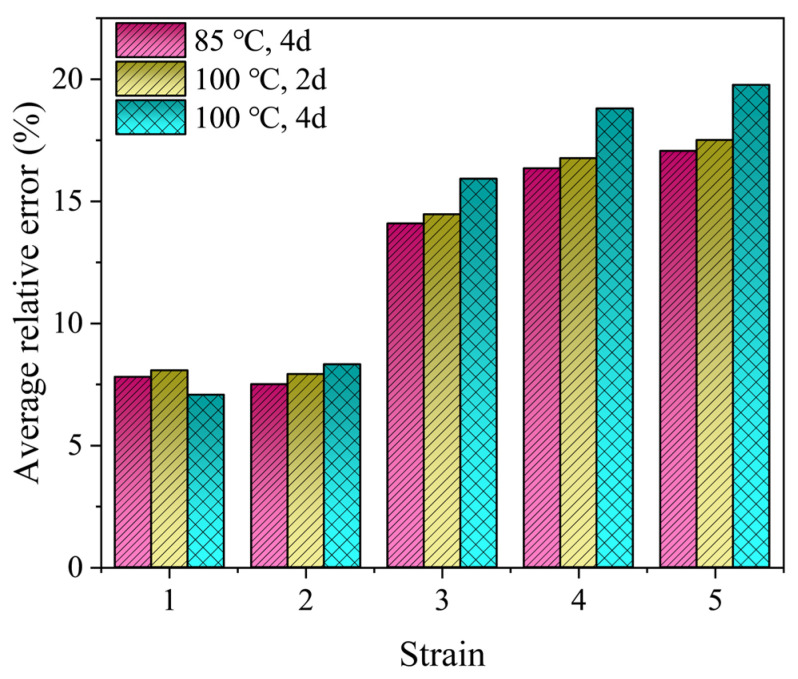
Average relative errors of the predicted results.

**Table 1 polymers-16-02608-t001:** Constitutive models and their stress and elongation relationships.

Model	Stress and Elongation Relationship	Parameters
Mooney–Rivlin	$\sigma =2C_{10}(\lambda -\lambda^{-2})+2C_{01}(1-\lambda^{-3})$	$C_{10},C_{01}$
Ogden (*N* = 3)	$\sigma =u_{1}(\lambda^{\alpha_{1}-1}-\lambda^{-\frac{1}{2}\alpha_{1}-1})+u_{2}(\lambda^{\alpha_{2}-1}-\lambda^{-\frac{1}{2}\alpha_{2}-1})+u_{3}(\lambda^{\alpha_{3}-1}-\lambda^{-\frac{1}{2}\alpha_{3}-1})$	*m*, *n*, *c*, *k* ^1^
Davis–De–Thomas	$\sigma =(m{\left(\lambda^{2}+\frac{2}{\lambda }-3+c^{2}\right)}^{-\frac{n}{2}}+4k(\lambda^{2}+\frac{2}{\lambda }-3))\cdot (\lambda -\lambda^{-2})$	*m*, *n*, *c*, *k*
modified Yeoh	$\sigma =(2C_{10}+4C_{20}(\lambda^{2}+\frac{2}{\lambda }-3)+6C_{30}{\left(\lambda^{2}+\frac{2}{\lambda }-3\right)}^{2}+2\alpha exp(-\beta (\lambda^{2}+\frac{2}{\lambda }-3)))\cdot (\lambda -\lambda^{-2})$	$C_{10},C_{20},C_{30},\alpha ,$ *β*
modified Gregory	$\sigma =A{{(\lambda }^{2}+\frac{2}{\lambda }-3+M^{2})}^{\alpha }+4B{\left(\lambda^{2}+\frac{2}{\lambda }-3+N^{2}\right)}^{\beta })\cdot (\lambda -\lambda^{-2}$)	*A*, *B*, *M*, *N*, α, *β*
gen-Yeoh	$\sigma =2(K_{1}\cdot m{\left(\lambda^{2}+\frac{2}{\lambda }-3\right)}^{m-1}+K_{2}\cdot p{\left(\lambda^{2}+\frac{2}{\lambda }-3\right)}^{p-1}+K_{3}\cdot q{\left(\lambda^{2}+\frac{2}{\lambda }-3\right)}^{q-1})\;(\lambda -\lambda^{-2})$	*K*_1_, *K*_2_, *K*_3_, *m*, *p*, *q*
Gregory	$\sigma =A{\left(\lambda^{2}+\frac{2}{\lambda }-3+C^{2}\right)}^{-\frac{n}{2}}+4B{\left(\lambda^{2}+\frac{2}{\lambda }-3+C^{2}\right)}^{\frac{m}{2}}$)$\cdot (\lambda -\lambda^{-2})$	*A*, *B*, *C*, *m*, *n*
Alexander	$\sigma =2(C_{1}exp(k{\left(\lambda^{2}+\frac{2}{\lambda }-3\right)}^{2})+\frac{1}{\lambda }(\frac{C_{2}}{2\lambda +\lambda^{-2}+\gamma -3}+C_{3}))\;\cdot (\lambda -\lambda^{-2})$	$C_{1},C_{2},C_{3},k,$ *γ*

## Data Availability

The original contributions presented in the study are included in the article, further inquiries can be directed to the corresponding author.

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
