# Peer review of "Constitutive Model for Thermal-Oxygen-Aged EPDM Rubber Based on the Arrhenius Law"

_polymers, 2024, doi:10.3390/polym16182608_

Round 1

Reviewer 1 Report

Comments and Suggestions for Authors

1.     The abstract lacks quantitative results, and the practical implications of the main findings are not highlighted at the end of the abstract.

2.     Please emphasize the significance and contribution of this study at the end of the introduction.

3.     The ISO standard (line#95) is not cited in the list of references.

4.     The accuracy of the equipment used is not mentioned in section 2.2. For example, the thermal aging chamber +/- 1 deg C, and tensile machine.

5.     The results presented in Figure 2 are not clearly mentioned, such as the average of 3 samples or a single sample.

6.     In section 2.3, the effect of the temperature and ageing duration on the stress strain plots is not discussed in-depth. For example, how does temperature influence the stress strain of the rubber?

7.     The words in Fig and Figure are not consistent in the manuscript (line#171, line#234).

8.     The deviation of results for Mooney-Rivlin is not analyzed and mentioned in section 3.3.

9.     The value of R-square is not provided in Figure 5 for all fitting curves. Please give the R-square in the graphs.

Author Response

Comment 1. The abstract lacks quantitative results, and the practical implications of the main findings are not highlighted at the end of the abstract.

Response:

Thanks for the proposal! We add the quantitative results of the research in the abstract and the practical implications of the main findings are highlighted at the end of the abstract. The added content is marked red in the revision manuscript. Please have a look, thanks!

Comment 2. Please emphasize the significance and contribution of this study at the end of the introduction.

Response:

Thanks for the proposal! We add some sentences to emphasize the significance and contribution of this study at the end of the introduction. The added content is marked red in section “1. Introduction” of the revision manuscript. Please have a look, thanks!

Comment 3.  The ISO standard (line#95) is not cited in the list of references.

Response:

Thanks for the proposal! We added the reference of the ISO standard (line#95) in the section of “References”. The added reference is marked red in the revision manuscript. Please have a look, thanks!

Comment 4.  The accuracy of the equipment used is not mentioned in section 2.2. For example, the thermal aging chamber +/- 1 deg C, and tensile machine.

Response:

Thanks for the proposal! The equipments in experiments are a thermal aging chamber and a tensile machine. The accuracy of the thermal aging chamber is ±1℃, and the tensile rate accuracy of the tensile machine is ±0.1%. We added those description in section “2.2 Experiment setup” of the revision manuscript, and the added content is marked red. Please have a look, thanks!

Comment 5.  The results presented in Figure 2 are not clearly mentioned, such as the average of 3 samples or a single sample.

Response:

Thanks for the proposal! The results presented in Figure 2 in the original manuscript (Figure 3 in the revision manuscript) are the median value of the three repeat samples, we add this statement in the “2.3 Experiment setup” section. The added sentence is marked red in the revision manuscript. Please have a look, thanks!

Comment 6.  In section 2.3, the effect of the temperature and ageing duration on the stress strain plots is not discussed in-depth. For example, how does temperature influence the stress-strain of the rubber?

Response:

谢谢你的提议!原稿中的图 2(修订稿中的图 3)分别显示了 EPDM 橡胶在 70°C、85°C 和 100 °C 下不同老化天数的拉伸应力-应变曲线。可以看出,随着老化时间的增加,应力在初始阶段迅速增加,然后缓慢增加,最后在应变超过一定值时再次迅速增加。EPDM 橡胶的拉伸性能表现出很强的非线性,尤其是在小变形下随着应变的增加而迅速变硬,这与一些高填充橡胶相似。此外,可以发现,随着温度的升高,小拉伸应变时应力-应变曲线的斜率略有增加,而当拉伸应变较大时,曲线的斜率显著增加。在 70°C 温度下老化 84 天,在 500% 应变下的应力为 8.52MPa,在 85°C 老化 84 天时,该值为 10.19MPa。当温度达到 100°C 时效 77 天时,500% 应变下的应力达到 13.27MPa,约为 70°C 时效 84 天的 1.6 倍。这表明温度在一定应变下会增强强度,并导致材料具有很强的非线性。所有这些观察到的现象都表明,温度和老化时间在机械性能的变化中都起着重要作用。

我们修改了“2.3 测试结果”部分,修改后的内容在修订稿件中标记为红色。请看一看,谢谢!

评论 7.图和图中的单词在手稿中不一致(行 #171、行 #234)。

响应:

谢谢你的提议!我们修改了图 1 中不一致的单词。和手稿中的图(第 #171 行、第 #234 行),同时,我们检查了整篇论文以避免同样的错误。请看一看,谢谢!

评论 8.Mooney-Rivlin 的结果偏差在第 3.3 节中没有分析和提及。

响应:

谢谢你的提议!使用 N=3 阶的 Mooney-Rivlin 模型和 Ogden 模型拟合未老化橡胶的实验应力-应变数据,如修订手稿中的图 4 所示。由此可见,Mooney-Rivlin 模型的拟合结果与拉伸试验数据存在显著偏差,模型无法拟合大变形下应力-应变曲线的“S 形”特征,这在以往的许多研究中都有发现[参考文献 [5] 和 [30] 在修订稿中]。从图 4 中我们可以看到,虽然 N=3 阶的 Ogden 模型可以反映大变形中拉伸行为的主要特征,但在小变形范围内,拟合曲线与测试数据之间仍然存在明显的偏差。Ogden 模型成功地捕获了多种橡胶的非线性超弹性行为,但它无法很好地拟合图 4 中的数据。这可能是由于在低应变范围内应力随应变的增加而迅速增加,并且应力-应变曲线表现出更突出的非线性特征。

我们在“3.3.Evaluation of the models for the EPDM rubber“部分,修改后的内容在修订稿中标记为红色。请看一看,谢谢!

评论 9.图 5 中并未提供所有拟合曲线的 R 平方值。请在图表中给出 R 平方。

响应:

谢谢你的提议!方程 (8) 用于通过非线性最小二乘法将图 6 中的数据拟合到修订稿中。在这里,我们采用非线性表面拟合方法,可以同时拟合三种温度下三条曲线的参数 m、n、c 和 k。拟合曲线如图 6 所示。老化模型参数 m、n、c 和 k 的 R 方值分别为 0.9679、0.9691、0.969 和 0.9059。我们在修改后的手稿中添加了拟合方法描述,并提供了图 6 中所有拟合曲线的 R 平方值。

我们修改了“3.4.温度和老化时间对模型参数的影响“部分,添加的内容在修订稿中标记为红色。图 6 也被修改了。请看一看,谢谢!

Reviewer 2 Report

Comments and Suggestions for Authors

The paper by Xiaoling Huis devoted to the study of thermal ageing of composite based on Ethylene-propylene-diene rubber. In this paper the authors proposed a new approach to the determination of temperature dependence of mechanical characteristics of composites after controlled ageing. Non-linear stress-strain curves were determined using physical-mechanical methods. Using physical chemistry approaches, a model was developed to predict the behaviour of the material at different oxidation times and temperatures. The work is a major contribution to the development of theoretical approaches for determining thermal oxidation resistance. The results of this study will optimise the production process and performance of polymer composites.

However, the manuscript needs some major revision in terms of writing. My specific comments are listed below.

1. Specify the errors of parameters m, n, c and k when approximating the temperature dependence. Why was the exponential dependence of approximation of these parameters chosen (equation 9)? Perhaps the Kohlrausch-Williams-Watts equation will show better results.

2. The text of the paper should explain in more detail why the function 𝑓(𝑥) is an exponentially function. It is common for chemical kinetics of heterogeneous reactions to be linear or power functions.

3. What may account for the strong difference between the modelled curves and the experimental curves in Figure 7 should be discussed in detail in the text.

4. It is desirable to include TG/DSC curves in an oxygen atmosphere for this composite. In order to determine how much degradation of the material has happened.  

Author Response

Comment 1. Specify the errors of parameters m, n, c and k when approximating the temperature dependence. Why was the exponential dependence of approximation of these parameters chosen (equation 9)? Perhaps the Kohlrausch-Williams-Watts equation will show better results.

Response:

Thanks for the proposal! The famous Kohlrausch-Williams-Watts (KWW) relaxation function or the stretched exponential relaxation function is an important observation in complex systems from the intricate behavior of liquids and glasses, the folding of proteins, to the structure and dynamics of atomic and molecular clusters, describing well the phenomena of important time-dependent dynamic processes. However, for rubber thermal ageing process, the rate of thermal degradation generally satisfies the Dakin-type kinetic relationship:

 dx/dt=kf(x) (1)

where x is the investigated property. Here, x represents the four model parameters m, n, c and k, respectively. t is the thermal aging time, k is the kinetic rate and f(x) is a function of the degree of material degradation. The kinetic rate k related to the temperature is assumed to follow the Arrhenius law:

k=Aexp(-Ea/RT) (2)

where A is the pre-exponential factor, Ea the activation energy, R the gas constant and T the absolute temperature. Because all of the parameters caused by thermal aging follow an exponential relation with aging time, the final f(x) for all parameters is defined as:

f(x)=exp(-ax) (3)

in which a is parameter which needs to be determined by fitting.

Combine the above three equations, the relationship functions between the four parameters and temperature and aging time can be written uniformly as:

x=1/aln(A exp(-Ea/RT) t+exp(ax0) )(4)

where  corressponding to the model parameters of the unaged rubber.

By fitting the data of the four model parameters under different aging times and temperatures, the temperature- and aging time-dependent model parameters were established as equation (9). The root-mean-square errors for the aging model parameters m, n, c and k are 0.03465, 0.00803, 0.00218 and 0.000656, respectively.

Comment 2. The text of the paper should explain in more detail why the function f(x) is an exponentially function. It is common for chemical kinetics of heterogeneous reactions to be linear or power functions.

Response:

Thanks for the proposal! In the research process of this work, we tried to use a linear function, that is, f(x)=x, to describe the change of the model parameters with aging time and aging temperature, and found that the linear function could not represent the change of each parameter well. We then tried the exponential function and polynomial, and found that the exponential function can achieve better fitting effect with few parameters, and finally we chose the exponential function. Thanks to your comment, we found that power functions can be used for chemical kinetic reactions in addition to linear functions. Therefore, we try to use the power function f(x)=xn to describe the change of parameters with aging time and aging temperature. We find that the power function can indeed describe the change of parameters well, and the function form is relatively simple. Therefore, the f(x) in the original paper is changed from exponential function to power function, and the final constitutive model of EPDM rubber considering thermal aging is derived.

We modified the “3.4. Effect of temperature and ageing time on model parameters” section, and the modified content is marked red in the revision manuscript. Please have a look, thanks!

Comment 3. What may account for the strong difference between the modelled curves and the experimental curves in Figure 7 should be discussed in detail in the text.

Response:

Thanks for the proposal!

In the original manuscript, we used three sets of experimental data (with aging time of 14 days, 28 days and 56 days, respectively) to conduct a preliminary test on the accuracy of the new model, and found that the model had a good fitting effect. The new test data with aging time of 2 days and 4 days were used to further test the prediction ability of the model. It is found that there is a large deviation between the model prediction results and the experimental results for large deformation. Prompted by this comment of the reviewer, we used the model to fit the stress-strain curves of three temperatures with aging time 7 days, and found that the fitting results were not as good as those with aging time 14 days, 28 days and 56 days. The fitting results were still good at small strains, but there were also large deviations in the stage of large deformation.

The aging model is based on the relationship between the parameters of the rubber Davis-De-Thomas model and aging time/temperature. Therefore, the accuracy of the relationship is particularly important for the prediction effect of aging model. Based on the rubber constitutive model, we analyzed the large deviation between the experimental results and the model results under the condition of short aging time and large deformation. Assuming that the strain ε approaches 0 (i.e. the tensile ratio λ approaches 1), the expression λ2+2/λ-3 in the model will approaches 0, so it is with the expression λ-λ-2 which can be assumed to be a very small amount ξ, then the stress given by the model can be estimated, σ ~ ξmc-n. So, at very small deformation, σ is only related to the parameters m, n and c, and is almost unaffected by k. However, when the deformation (tensile ratio λ) gradually increases, the coefficient of k given by the model is 4(λ2+2/λ-3)(λ-λ-2). This coefficient increases exponentially with the increase of the tensile ratio λ, and the result of it is shown in Figure 2 below.

Figure 2. The result of the coefficient of k as a function of the tensile ratio λ.

Therefore, with the increase of deformation, the stress σ of the model will gradually increase under the influence of k. Based on the fitting results of the four parameters, the R-square values of parameters m, n and c are 0.9679, 0.9691 and 0.969, respectively, while the R-square value of parameter k is the lowest one which is only 0.9059. As can be seen from Figure 6 (for parameter k) in the revision manuscript, the relative fitting deviation under short aging time is larger than that under long aging time, which is the reason that the model has a large deviation in the large strain range when predicting the stress-strain curve with short aging time. This study aims to provide a method to construct a constitutive model of thermo-oxygen aging. Thank you very much for your constructive comment, In the future research work, we will do detailed experiments, carry out high-precision fitting, and make the prediction results of the theoretical model more accurate.

We modified the “3.5. Model verification” section based on the above analysis, and the added content is marked red in the revision manuscript. Please have a look, thanks!

评论 4.对于这种复合材料,最好在氧气气氛中包含 TG/DSC 曲线。为了确定材料发生了多少降解。

响应:

谢谢你的提议!TG/DSC 测试方法确实是测量材料受老化影响程度的好方法。但是,在老化后 16 到 24 小时内对老化样品进行检测效果更好,而且我们的老化时间相对较长,很难用 TG/DSC 方法研究此时老化条件对样品降解程度的影响。非常感谢您的建议。在未来与橡胶老化相关的研究工作中,我们将使用 TG/DSC 方法系统研究老化对材料降解程度的影响。

Round 2

Reviewer 2 Report

Comments and Suggestions for Authors

The authors of the paper listened to my comments and made the necessary corrections to the work. I recommend this paper for acceptance.